# Antecedents of loneliness among cancer survivors: An exploratory analysis of the Health Information National Trends Survey (HINTS) data

Avishek Choudhury[1]*, Christopher W. Wheldon[2], Safa Elkefi[3], Priyanka Dixit[4]

1 West Virginia University, Morgantown, West Virginia, United States of America, 2 Temple University, Philadelphia, Pennsylvania, United States of America, 3 State University of New York, Binghamton, New York, United States of America, 4 TATA Institute of Social Sciences, Mumbai, Maharashtra, India

* avishek.choudhury@mail.wvu.edu

## Abstract

Social isolation is a growing public health concern, particularly among cancer survivors who face persistent challenges in maintaining social connections following treatment. While its impact on mental health is increasingly recognized, the underlying pathways and contextual differences across rural and urban settings remain underexplored. This study aimed to identify the psychosocial and personal factors that contribute to perceived social isolation among U.S. cancer survivors and to assess how isolation mediates the relationship between these factors and mental health. We also examined whether these associations differ between rural and urban populations. Data were drawn from the 2022 Health Information National Trends Survey (HINTS 6), with a subsample of cancer survivors (n = 926). A conceptual framework was developed using constructs from the Biopsychosocial Model, the Stress Process Model, the Transactional Model of Stress and Coping, and the Andersen Behavioral Model of Health Services Use. Partial Least Squares Structural Equation Modeling (PLS-SEM) was used to evaluate the relationships among personal factors (age, BMI, time since diagnosis, inability of self-care), psychosocial perceptions (perceived care quality, cancer information comprehension, negative life perception), social isolation, and mental health (PHQ-4). Multigroup analysis compared rural and urban survivors. Social isolation was a strong predictor of mental health, with key antecedents including cancer information access, inability of self-care, time since diagnosis, negative life perception, and perception of care quality. Several pathways varied by geography; for instance, the effects of self-care difficulty and care perception on isolation and mental health were significant only in urban settings. This study highlights the central role of social isolation in shaping mental health outcomes among cancer survivors and underscores the importance of targeted, context-sensitive interventions to reduce isolation and promote psychosocial well-being, particularly in underserved rural communities.

**Data availability statement:** The datasets generated during and/or analyzed during the current study is available upon request to NCI HINTS at their website https://hints.cancer.gov/data/download-data.aspx. Authors do not have the rights to share the data in any form.

**Funding:** The author(s) received no specific funding for this work.

**Competing interests:** The authors have declared that no competing interests exist.

## Introduction

Social isolation is increasingly recognized as a critical issue affecting cancer survivors [1]. The U.S. Surgeon General identified social isolation as significant threats to individual and population health [2] but emerging evidence suggests that these issues may be particularly detrimental for cancer survivors [3]. The National Academies of Sciences, Engineering, and Medicine (NASEM), in their report Social Isolation or Loneliness in Older Adults: Opportunities for the Health Care System, highlight isolation among cancer survivors as a pressing public health concern, calling for increased attention to this issue at a national level [4]. While the cancer experience itself can be isolating, social isolation often intensifies over time due to persistent physical limitations, changes in social roles, and weakened support networks. Cancer-related fatigue, cognitive impairments, and other long-term treatment effects can limit participation in social activities, leading to further social withdrawal [1].

Emerging evidence suggests that social isolation may contribute to increased mortality risk among cancer survivors. Zhao et al., found that elevated isolation was significantly associated with a higher risk of mortality in this population, adding to growing evidence that social disconnection may directly influence cancer outcomes [5]. Hyland et al. demonstrated that among lung cancer patients, social isolation was strongly correlated with depressive symptoms and reduced quality of life, potentially mediated by social constraints and stigma [6]. It has also been linked to biological mechanisms relevant to cancer progression, such as heightened inflammatory responses and dysregulated stress pathways [7]. Choudhury in 2023 noted a quadratic effect of isolation on mental health, with higher levels of isolation associated with worse mental health outcomes [8]. A scoping review by Pilleron et al. (2023) found that loneliness is prevalent among older adults with cancer and may increase during the first 6–12 months after diagnosis [9]. Furthermore, Wheldon et al. reported a 70% increased risk of moderate to severe loneliness in cancer survivors more than five years post-diagnosis [1].

Although isolation is increasingly recognized as a health concern among older adults, few studies have examined how its predictors and health correlates may vary by geographic context. Henning-Smith and colleagues, using data from the National Social Life, Health, and Aging Project, found that the adjusted correlates of social isolation differed by rurality [10]. For example, physical health was significantly associated with isolation among urban residents, while other factors such as employment and race showed unique associations in micropolitan and noncore rural settings. However, the study was limited in scope, relying on data collected in 2010–2011 and focusing on a narrow range of health indicators. Cancer-related outcomes, mental health symptoms, and behavioral health risks were not included, despite their known relevance to survivor well-being. Given the rapidly changing social and health environments faced by today's cancer survivors, updated evidence is needed.

The present study seeks to address these gaps by identifying risk factors for loneliness and its health correlates among U.S. cancer survivors, across rural and urban populations. Additionally, we also explore the relationship between social isolation and mental health. Fig 1 shows the conceptual framework explored in this study. The

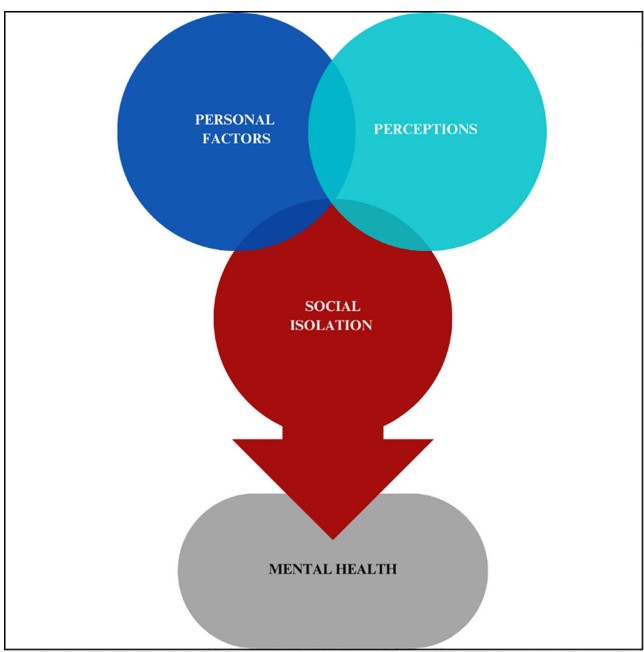

**Fig 1. Conceptual framework illustrating the influence of personal factors and psychosocial perceptions on mental health among cancer survivors, with social isolation as a central mediating pathway.**

framework was derived from the Biopsychosocial Model, the Stress Process Model, the Transactional Model of Stress and Coping, and the Andersen Behavioral Model of Health Services Use. The development of this framework has been described in the following section.

## Method

This study is a secondary data analysis using publicly available, de-identified data from the Health Information National Trends Survey. As the dataset does not contain personally identifiable information and is freely accessible for research purposes, Institutional Review Board approval was not required.

### Data source

We used the Health Information National Trends Survey (HINTS) data. HINTS is a nationally representative survey conducted by the National Cancer Institute (NCI) to assess the public's access to, use of, and perceptions regarding health information. Within HINTS, we used HINTS 6 (2022; n = 6,252) data.

### Conceptual framework development

The development of our conceptual framework (Fig 1) was informed by an integration of established theoretical models in health psychology and behavioral medicine, alongside an iterative, data-driven process using the HINTS. We focused specifically on cancer patients and survivors (n = 926).

Our framework draws primarily from three theoretical models. First, the Biopsychosocial Model provided a foundational perspective, recognizing that health is shaped not only by biological and clinical factors but also by psychosocial influences [11]. This model supported the inclusion of a multidimensional set of predictors that extend beyond disease status. Second, the Stress Process Model informed our decision to include mediating constructs, specifically perceived social

isolation (i.e., loneliness), that may help explain how upstream factors such as physical limitations or negative life perceptions translate into mental health outcomes [12]. This model emphasizes the accumulation of stressors over time and the role of social and psychological resources in buffering or exacerbating their effects. Third, we incorporated elements from the Transactional Model of Stress and Coping, which posits that individuals' cognitive appraisal of life events (e.g., cancer diagnosis and survivorship) shapes their emotional and behavioral responses [13]. This model supported the inclusion of subjective perceptions, such as how survivors assess their quality of care, life outlook, and understanding of cancer information, as core predictors of emotional well-being. Finally, the Andersen Behavioral Model of Health Services Use provided a structural lens for understanding how access to and perception of health services, particularly cancer-related information and care quality, can influence both care utilization and psychological outcomes [14]. Perceived quality of care and health information access are conceptualized as enabling resources that influence health behavior and outcomes.

Guided by these models, we initially selected a broad set of candidate variables available in the HINTS dataset across three conceptual domains. We explored a range of demographic and health-related characteristics, including age, gender, race/ethnicity, educational attainment, income, insurance status, comorbidities, time since diagnosis (TDx), body mass index (BMI), and inability of self-care (ISC). We considered variables related to emotional well-being, perceived control, optimism, trust in healthcare providers, perceived quality of care (PCQ), negative life perception (NLP), and cancer information access and comprehension (CIAC). We assessed social support, frequency of social contact, and perceived social isolation (SI).

Using structural equation modeling (SEM), we tested multiple iterations of the conceptual model. Our goal was to identify a parsimonious model that maintained theoretical integrity while demonstrating acceptable model fit and retaining statistically significant paths. Variables that were highly collinear, inconsistently associated with key outcomes, or contributed minimal explanatory power were excluded in subsequent iterations. For example, although education and income are well-known social determinants of health, their indirect effects on mental health through social isolation or perceptions were non-significant in our initial models and were therefore removed. Similarly, trust in providers was conceptually relevant but did not yield strong or consistent pathways in the model when compared to PCQ and NLP, which better captured the survivor's experience of care and life outlook.

Through this iterative process, we identified a final set of variables that best captured the key mechanisms linking survivorship experiences to mental health outcomes. As shown in Table 1 some variables were used to form latent construct when appropriate.

## Statistical analyses

We employed Partial Least Squares Structural Equation Modeling (PLS-SEM) to examine the relationships between latent constructs and their observed indicators. PLS-SEM is a variance-based structural equation modeling technique suitable for complex models with multiple dependent and independent variables. It allows for the simultaneous evaluation of the measurement model, which assesses construct reliability and validity, and the structural model, which tests hypothesized relationships between constructs. Additionally, PLS-SEM is well suited for exploratory analyses, it is robust against non-normal data distributions and small sample sizes.

The PLS algorithm is structured as a sequence of regressions in terms of weight vectors. These weight vectors, upon convergence, satisfy fixed point equations. The core PLS algorithm involves three main stages: iterative estimation of latent variable scores, estimation of outer weights/loadings and path coefficients, and estimation of location parameters [15]. The first stage follows a four-step iterative procedure: outer approximation of latent variable scores, estimation of inner weights, inner approximation of latent variable scores, and estimation of outer weights. These steps are repeated until convergence is achieved. The second stage estimates the outer weights/loadings for each indicator and the path coefficients for the structural model. The final stage estimates the intercept terms for the latent variables in the model.

**Table 1. The survey instrument.**

| Questions (observed variables) | Latent constructs | Participant Responses |
|---|---|---|
| Based on the results of your most recent search for information about cancer, how much do you agree or disagree: You were concerned about the quality of the information: *(CancerConcernedQuality)* | Cancer Information Access and Comprehension (CIAC) | 1 = Strongly agree; 2 = Somewhat agree; 3 = Somewhat disagree; 4 = Strongly disagree; |
| Based on the results of your most recent search for information about cancer, how much do you agree or disagree: You felt frustrated during your search for the information: *(CancerFrustrated)* | | |
| Based on the results of your most recent search for information about cancer, how much do you agree or disagree: It took a lot of effort to get the information you needed: *(CancerLotOfEffort)* | | |
| Based on the results of your most recent search for information about cancer, how much do you agree or disagree: The information you found was hard to understand: *(CancerHardToUnderstand)* | | |
| In the past 12 months, how often did your doctors, nurses, or other health professionals give you the chance to ask all the health-related questions you had? *(ChanceAskQuestions)* | Perception of Care Quality (PCQ) | 1 = Always; 2 = Usually; 3 = Sometimes; 4 = Never; |
| In the past 12 months, how often did your doctors, nurses, or other health professionals explain things in a way you could understand? *(ExplainedClearly)* | | |
| In the past 12 months, how often did your doctors, nurses, or other health professionals give the attention you needed to your feelings and emotions? *(FeelingsAddressed)* | | |
| In the past 12 months, how often did your doctors, nurses, or other health professionals involve you in decisions about your health care as much as you wanted? *(InvolvedDecisions)* | | |
| In the past 12 months, how often did your doctors, nurses, or other health professionals spend enough time with you? *(SpentEnoughTime)* | | |
| In the past 12 months, how often did your doctors, nurses, or other health professionals make sure you understood the things you needed to do to take care of your health? *(UnderstoodNextSteps)* | | |
| I have a clear sense of direction in life: *(ClearSenseDir)* | Negative Life Perception (NLP) | 1 = Very much; 2 = Quite a bit; 3 = Somewhat; 4 = A little bit; 5 = Not at all; |
| I experience deep fulfillment in my life: *(DeepFulfillment)* | | |
| My life has meaning: *(LifeHasMeaning)* | | |
| My life has purpose: *(LifeHasPurpose)* | | |
| I feel isolated from others: *(FeelIsolated)* | Social Isolation (SI) | 1 = Never; 2 = Rarely; 3 = Sometimes; 4 = Usually; 5 = Always; |
| I feel left out: *(FeelLeftOut)* | | |
| I feel that people barely know me: *(FeelPeopleBarelyKnow)* | | |
| I feel that people are around me but not with me: *(FeelPeopleNotWithMe)* | | |
| Over the past 2 weeks, how often have you been bothered by: Feeling down, depressed or hopeless? *(Hopeless)* | Mental Health (MH) | 1 = Nearly every day; 2 = More than half the days; 3 = Several days; 4 = Not at all; |
| Over the past 2 weeks, how often have you been bothered by: Little interest or pleasure in doing things? *(LittleInterest)* | | |
| Over the past 2 weeks, how often have you been bothered by: Feeling nervous, anxious or on edge? *(Nervous)* | | |
| Over the past 2 weeks, how often have you been bothered by: Not being able to stop or control worrying? *(Worrying)* | | |
| **Single items questions** | **Variable name** | **Participant Responses** |
| Overall, how confident are you about your ability to take good care of your health? *(SelfCareInability)* | Inability of Self Care (ISC) | 1 = Completely confident; 2 = Very confident; 3 = Somewhat confident; 4 = A little confident; 5 = Not confident at all; |
| How long ago were you diagnosed with cancer? *(TimeSinceDX)* | Time Since Dx (TDx) | 1 = Less than 1 Yr Since DX; 2 = 2–5 Yrs Since DX; 3 = 6–10 Yrs Since DX; 4 = 11 + Yrs Since DX; |

*(Continued)*

**Table 1.** (Continued)

| Questions (observed variables) | Latent constructs | Participant Responses |
|---|---|---|
| Body Mass Index (BMI; Derived variable by HINTS) | BMI | N/A |
| Age Group (Age; Derived variable by HINTS) | Age | 1 = 18-34;<br>2 = 35-49;<br>3 = 50-64;<br>4 = 65-74;<br>5 = 75 and older; |

We used the path weighting scheme which generally results in the highest R² values for endogenous latent variables. Additionally, PLS weighting vectors were applied, and weights provided by the HINTS data were used to ensure representative and unbiased estimates. Missing values were handled using the pairwise deletion method, allowing the retention of as much data as possible while excluding only the missing values for specific analyses. Initial outer weights were set to 1 for all indicators. The maximum number of iterations was fixed at 3,000, and the stop criterion was set at $10^{-7}$, ensuring the algorithm stopped when the change in outer weights between consecutive iterations fell below this threshold.

The measurement model consisted of four reflective latent constructs, each measured using multiple observed indicators. Reliability was assessed based on factor loadings greater than 0.70. Internal consistency reliability and validity were evaluated using composite reliability (rhoC), with a threshold greater than 0.70, and the average variance extracted (AVE), which needed to exceed 0.50.

Following the validation of the measurement model, bootstrapping with 10,000 iterations was conducted to obtain bias-corrected standardized parameter estimates, p-values, and confidence intervals.

For exploratory purposes, we also examined potential non-linear relationships among key variables by testing quadratic effects within the structural model. This approach allowed us to assess whether the influence of certain predictors on outcomes, such as mental health, changed in magnitude or direction at different levels of the predictor. While several quadratic terms were initially tested, only those that reached statistical significance were retained in the final model to ensure parsimony and interpretability.

Lastly, to explore potential differences in the relationships among constructs based on geographic location, we conducted a multigroup analysis by dividing the dataset into two groups: participants from urban regions and those from rural areas, as categorized by HINTS. We then applied the same PLS-SEM model to both groups separately and compared the results to determine whether the significant predictors remained consistent across both models.

## Result

The dataset consisted of 6252 respondents, of which about 14.81% (n = 926) were diagnosed with cancer. Table 2 shows the participant characteristics.

Table 3 shows the data description of the factors used in our study.

Fig 2 shows the Spearman rank-order correlation coefficients indicating the interconnectedness of psychological, social, and healthcare-related factors in shaping individuals' experiences and perceptions. Several strong positive correlations are evident among related constructs, such as *Cancer Concerned Quality*, *Cancer Frustrated*, and *Cancer Lot of Effort*, suggesting that individuals who expressed concern about the quality of cancer information also reported frustration and effort in finding information. Similarly, variables related to healthcare experiences, such as *Explained Clearly*, *Feelings Addressed*, and *Involved Decisions*, exhibit high positive correlations, indicating that positive experiences in one aspect

**Table 2. Participant characteristics.**

| Variables | Urban | Rural |
|---|---|---|
| | Frequency (%) | Frequency (%) |
| **Sex** | | |
| *Male* | 469 (59.90) | 84 (58.74) |
| *Female* | 314 (40.10) | 59 (41.26) |
| **Marital status** | | |
| *Married* | 376 (48.02) | 74 (51.75) |
| *Living as married or living with a romantic partner* | 22 (2.81) | 7 (4.90) |
| *Divorced* | 142 (18.14) | 25 (17.48) |
| *Widowed* | 140 (17.88) | 28 (19.58) |
| *Separated* | 16 (2.04) | 4 (2.80) |
| *Single, never been married* | 87 (11.11) | 5 (3.50) |
| **Education** | | |
| *Less than High School* | 45 (5.75) | 11 (7.69) |
| *High School Graduate* | 114 (14.56) | 38 (26.57) |
| *Some Colleges* | 219 (27.97) | 50 (34.97) |
| *Bachelor's Degree* | 191 (24.39) | 25 (17.48) |
| *Post-Baccalaureate Degree* | 181 (23.21) | 15 (10.49) |
| *Not reported* | 33 (4.21) | 4 (2.80) |
| **Age (years)** | | |
| *18-34* | 8 (1.02) | 1 (<1.0) |
| *35-49* | 50 (6.39) | 8 (5.59) |
| *50-64* | 199 (25.42) | 37 (25.87) |
| *65-74* | 262 (33.46) | 58 (40.56) |
| *75 and older* | 250 (31.93) | 36 (25.17) |
| *Not reported* | 14 (1.79) | 3 (2.10) |
| **Race** | | |
| *Non-Hispanic White* | 518 (66.16) | 108 (75.52) |
| *Non-Hispanic Black or African American* | 85 (10.86) | 7 (4.90) |
| *Hispanic* | 69 (8.81) | 2 (1.40) |
| *Non-Hispanic Asian* | 18 (2.30) | 0 (0) |
| *Non-Hispanic Other* | 17 (2.17) | 6 (4.20) |
| *Not reported* | 76 (9.71) | 20 (13.99) |

of patient-provider communication are likely associated with positive experiences in others. Conversely, strong negative correlations are observed between measures of social isolation (e.g., *Feel Isolated*, *Feel Left Out*) and measures of life perception (e.g., *Life Has Meaning*, *Life Has Purpose*). Mental health indicators like *Hopelessness* and *Worrying* show strong negative associations with positive psychological constructs, reinforcing the expected inverse relationship between mental distress and well-being.

## Measurement model

The results in Table 4 provide evidence of convergent validity and reliability for the latent constructs included in the study. All observed variables exhibit high factor loadings (≥0.70), indicating strong associations with their respective latent constructs. The AVE values exceed the recommended threshold of 0.50, confirming that each construct captures a sufficient

**Table 3. Data description.**

| Observed variables | Frequency (%) | Mean (SD) |
|---|---|---|
| **Cancer Concerned Quality** | | 2.56 (1.03) |
| *Strongly agree* | 108 (17.45) | |
| *Somewhat agree* | 205 (33.12) | |
| *Somewhat disagree* | 160 (25.85) | |
| *Strongly disagree* | 146 (23.59) | |
| **Cancer Frustrated** | | 2.89 (0.97) |
| *Strongly agree* | 54 (8.77) | |
| *Somewhat agree* | 164 (26.62) | |
| *Somewhat disagree* | 191 (20.63) | |
| *Strongly disagree* | 207 (22.35) | |
| **Cancer Lot of Effort** | | 2.77 (0.96) |
| *Strongly agree* | 62 (6.70) | |
| *Somewhat agree* | 183 (19.76) | |
| *Somewhat disagree* | 209 (33.66) | |
| *Strongly disagree* | 167 (26.89) | |
| **Cancer Hard to Understand** | | 2.93 (0.92) |
| *Strongly agree* | 30 (6.51) | |
| *Somewhat agree* | 160 (26.06) | |
| *Somewhat disagree* | 215 (35.02) | |
| *Strongly disagree* | 199 (32.41) | |
| **Chance Ask Questions** | | 1.50 (0.68) |
| *Always* | 515 (59.61) | |
| *Usually* | 276 (31.94) | |
| *Sometimes* | 65 (7.52) | |
| *Never* | 8 (<1) | |
| **Explained Clearly** | | 1.46 (0.62) |
| *Always* | 526 (60.95) | |
| *Usually* | 279 (32.33) | |
| *Sometimes* | 57 (6.60) | |
| *Never* | 1 (<1) | |
| **Feelings Addressed** | | 1.75 (0.81) |
| *Always* | 393 (45.70) | |
| *Usually* | 307 (35.70) | |
| *Sometimes* | 141 (16.40) | |
| *Never* | 19 (2.21) | |
| **Involved Decisions** | | 1.60 (0.75) |
| *Always* | 467 (54.24) | |
| *Usually* | 286 (33.22) | |
| *Sometimes* | 93 (10.80) | |
| *Never* | 15 (1.75) | |
| **Spent Enough Time** | | 1.76 (0.83) |
| *Always* | 396 (46.21) | |
| *Usually* | 300 (35.01) | |
| *Sometimes* | 135 (15.75) | |
| *Never* | 26 (3.03) | |

*(Continued)*

| Observed variables | Frequency (%) | Mean (SD) |
|---|---|---|
| **Understood Next Steps** | | 1.52 (0.69) |
| *Always* | 507 (58.68) | |
| *Usually* | 274 (31.71) | |
| *Sometimes* | 75 (8.68) | |
| *Never* | 8 (<1) | |
| **Clear Sense Dir** | | 1.74 (0.93) |
| *Very much* | 468 (51.94) | |
| *Quite a bit* | 251 (27.86) | |
| *Somewhat* | 137 (15.21) | |
| *A little bit* | 34 (3.77) | |
| *Not at all* | 11 (1.22) | |
| **Deep Fulfillment** | | 2.02 (1.05) |
| *Very much* | 349 (38.86) | |
| *Quite a bit* | 285 (31.74) | |
| *Somewhat* | 182 (20.27) | |
| *A little bit* | 57 (6.35) | |
| *Not at all* | 25 (2.78) | |
| **Life Has Meaning** | | 1.57 (0.84) |
| *Very much* | 557 (61.75) | |
| *Quite a bit* | 208 (23.06) | |
| *Somewhat* | 111 (12.31) | |
| *A little bit* | 20 (2.22) | |
| *Not at all* | 6 (<1) | |
| **Life Has Purpose** | | 1.71 (0.96) |
| *Very much* | 504 (56.06) | |
| *Quite a bit* | 217 (24.14) | |
| *Somewhat* | 123 (13.68) | |
| *A little bit* | 45 (5.01) | |
| *Not at all* | 10 (1.11) | |
| **Feel Isolated** | | 1.88 (1.07) |
| *Never* | 447 (49.89) | |
| *Rarely* | 215 (24.00) | |
| *Sometimes* | 153 (17.08) | |
| *Usually* | 60 (6.70) | |
| *Always* | 21 (2.34) | |
| **Feel Left Out** | | 1.96 (0.96) |
| *Never* | 348 (38.75) | |
| *Rarely* | 301 (33.52) | |
| *Sometimes* | 197 (21.94) | |
| *Usually* | 38 (4.23) | |
| *Always* | 14 (1.56) | |
| **Feel People Barely Know Me** | | 1.99 (1.06) |
| *Never* | 376 (42.11) | |
| *Rarely* | 256 (28.67) | |
| *Sometimes* | 180 (20.16) | |
| *Usually* | 58 (6.49) | |
| *Always* | 23 (2.58) | |

*(Continued)*

**Table 3.** (Continued)

| Observed variables | Frequency (%) | Mean (SD) |
|---|---|---|
| **Feel People Not with Me** | | 1.86 (1.03) |
| *Never* | 439 (49.00) | |
| *Rarely* | 226 (25.22) | |
| *Sometimes* | 162 (18.08) | |
| *Usually* | 51 (5.69) | |
| *Always* | 18 (2.01) | |
| **Hopeless** | | 3.58 (0.74) |
| *Nearly every day* | 26 (2.89) | |
| *More than half the days* | 58 (6.45) | |
| *Several days* | 183 (20.36) | |
| *Not at all* | 632 (70.30) | |
| **Little Interest** | | 3.44 (0.88) |
| *Nearly every day* | 48 (5.32) | |
| *More than half the days* | 90 (9.98) | |
| *Several days* | 180 (19.96) | |
| *Not at all* | 584 (64.75) | |
| **Nervous** | | 3.48 (0.80) |
| *Nearly every day* | 39 (4.34) | |
| *More than half the days* | 58 (6.46) | |
| *Several days* | 230 (25.61) | |
| *Not at all* | 571 (63.59) | |
| **Worrying** | | 3.51 (0.79) |
| *Nearly every day* | 40 (4.44) | |
| *More than half the days* | 49 (5.44) | |
| *Several days* | 225 (25.00) | |
| *Not at all* | 586 (65.11) | |
| **Self-Care Inability** | | 2.11 (0.90) |
| *Completely confident* | 248 (27.28) | |
| *Very confident* | 381 (41.91) | |
| *Somewhat confident* | 230 (25.30) | |
| *A little confident* | 36 (3.96) | |
| *Not confident at all* | 14 (1.54) | |
| **Time Since DX** | | 3.08 (1.06) |
| *Less than 1 Yr Since DX* | 88 (10.22) | |
| *2-5 Yrs Since DX* | 184 (21.37) | |
| *6-10 Yrs Since DX* | 157 (18.23) | |
| *11 + Yrs Since DX* | 432 (50.17) | |
| **Age (years)** | | 3.90 (0.95) |
| 18-34 | 9 (<1) | |
| 35-49 | 58 (6.38) | |
| 50-64 | 236 (25.96) | |
| 65-74 | 320 (35.20) | |
| 75 and older | 286 (31.46) | |
| **BMI** | n/a | 28.25 (6.08) |

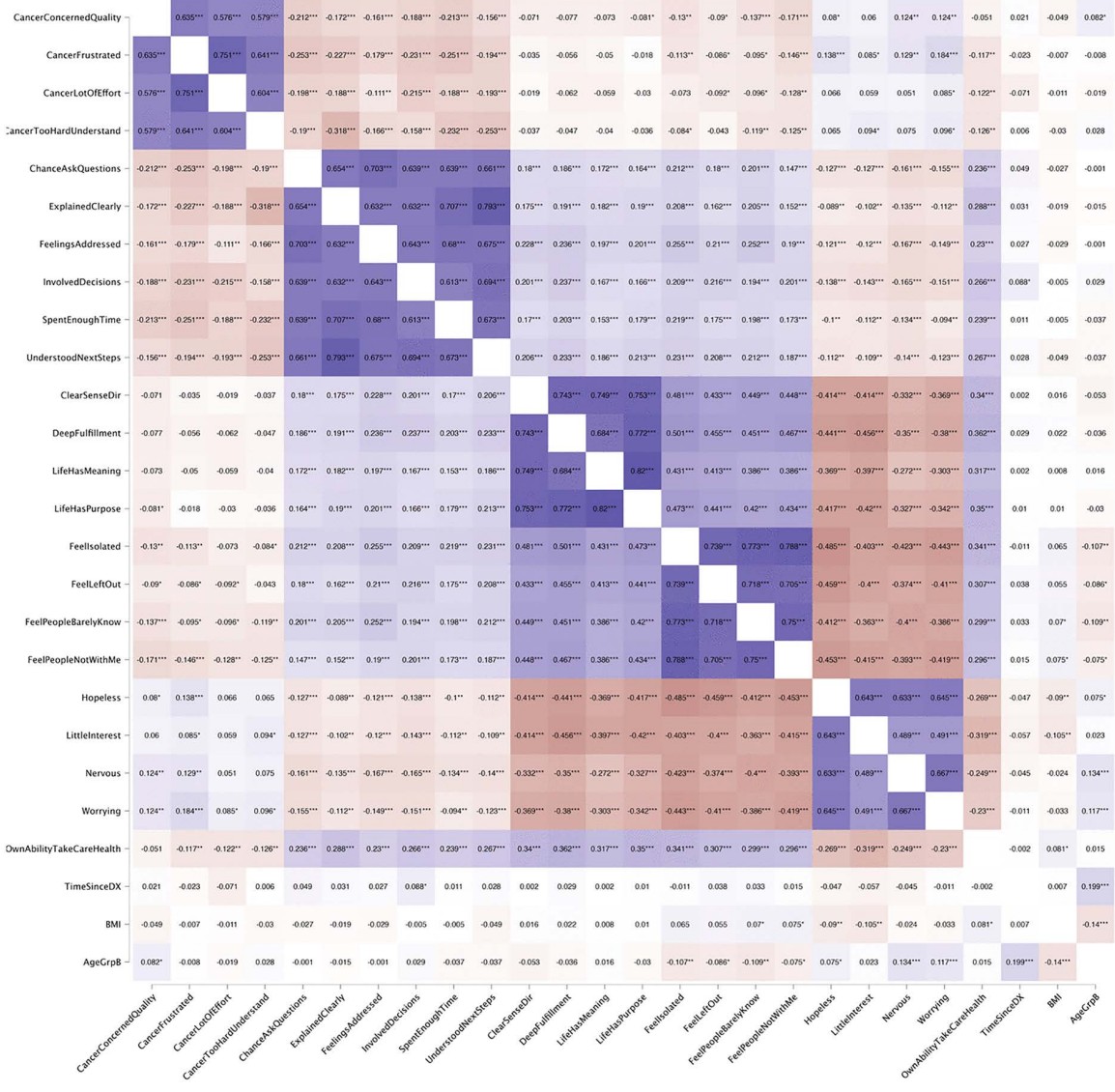

**Fig 2. The Spearman rank-order correlation coefficient.** The color gradient indicates positive (blue) and negative (red) correlations, with darker shades representing stronger associations.

proportion of variance from its indicators. Reliability measures demonstrate strong internal consistency. Composite reliability (Rho C) values range from 0.90 to 0.96, exceeding the acceptable threshold of 0.70, suggesting high reliability. Similarly, Dijkstra-Henseler's Rho (Rho A) and Cronbach's alpha values are consistently above 0.80, reinforcing the stability of the constructs.

## Structural model

The path analysis results as shown in Table 5 provide insights into the direct, indirect, and total effects of key variables on mental health (MH), social isolation (SI), body mass index (BMI), and other latent constructs. Age showed no significant direct effect on mental health ($\beta = 0.06$, $p = 0.26$) but had a significant total indirect effect ($\beta = 0.11$, $p < 0.001$), resulting in a

**Table 4. Convergent validity and reliability.**

| Questions (observed variables) | Factor loading | Latent constructs | AVE | Rho C | Rho A | Cronbach's Alpha |
|---|---|---|---|---|---|---|
| *Cancer Concerned Quality* | 0.81 | Cancer Information Access and Comprehension | 0.70 | 0.90 | 0.86 | 0.86 |
| *Cancer Frustrated* | 0.88 | | | | | |
| *Cancer Lot of Effort* | 0.83 | | | | | |
| *Cancer Hard to Understand* | 0.81 | | | | | |
| *Chance Ask Questions* | 0.82 | Perception of Care Quality | 0.71 | 0.94 | 0.92 | 0.92 |
| *Explained Clearly* | 0.87 | | | | | |
| *Feelings Addressed* | 0.81 | | | | | |
| *Involved Decisions* | 0.83 | | | | | |
| *Spent Enough Time* | 0.86 | | | | | |
| *Understood Next Steps* | 0.85 | | | | | |
| *Clear Sense Dir* | 0.92 | Negative Life Perception | 0.84 | 0.96 | 0.94 | 0.94 |
| *Deep Fulfillment* | 0.91 | | | | | |
| *Life Has Meaning* | 0.91 | | | | | |
| *Life Has Purpose* | 0.93 | | | | | |
| *Feel Isolated* | 0.91 | Social Isolation | 0.83 | 0.95 | 0.93 | 0.93 |
| *Feel Left Out* | 0.89 | | | | | |
| *Feel People Barely Know* | 0.91 | | | | | |
| *Feel People Not with Me* | 0.92 | | | | | |
| *Hopeless* | 0.90 | Mental Health | 0.74 | 0.92 | 0.90 | 0.88 |
| *Little Interest* | 0.75 | | | | | |
| *Nervous* | 0.89 | | | | | |
| *Worrying* | 0.88 | | | | | |
| *Self-Care Inability* | 1 | Inability of Self Care | 1 | 1 | 1 | 1 |
| *Time Since DX* | 1 | Time Since DX | 1 | 1 | 1 | 1 |
| BMI | 1 | BMI | 1 | 1 | 1 | 1 |
| Age | 1 | Age Group | 1 | 1 | 1 | 1 |

significant total effect ($\beta = 0.16$, $p < 0.001$), indicating that its impact on mental health operates through mediating variables. Age was also significantly negatively associated with social isolation ($\beta = -0.19$, $p < 0.001$) and BMI ($\beta = -0.05$, $p = 0.01$). BMI did not significantly predict mental health ($\beta = -0.04$, $p = 0.43$), suggesting that weight status alone may not be a strong determinant of mental well-being.

Cancer Information Access and Comprehension (CIAC) was negatively associated with Inability of Self-Care (ISC) ($\beta = -0.12$, $p = 0.05$) and Perception of Care Quality (PCQ) ($\beta = -0.34$, $p < 0.001$). Additionally, CIAC had significant negative total effects on social isolation ($\beta = -0.18$, $p < 0.001$) and negative life perception ($\beta = -0.08$, $p < 0.001$), suggesting that better cancer information access may contribute to improved psychological and social well-being. ISC significantly predicted negative life perception ($\beta = 0.39$, $p < 0.001$) and social isolation ($\beta = 0.35$, $p < 0.001$), reinforcing the notion that difficulties in self-care are linked to greater social disconnection and negative life outlook.

Negative life perception (NLP) was strongly associated with social isolation ($\beta = 0.47$, $p < 0.001$), BMI ($\beta = 0.12$, $p < 0.001$), and poorer mental health ($\beta = -0.26$, $p < 0.001$), emphasizing the critical role of perceived life satisfaction in overall well-being. PCQ positively influenced ISC ($\beta = 0.26$, $p < 0.001$) and negative life perception ($\beta = 0.10$, $p < 0.001$), while its total effect on mental health was negative ($\beta = -0.08$, $p < 0.001$), suggesting that better perceptions of care quality might improve certain aspects of well-being but do not necessarily protect against mental distress.

**Table 5. Standardized direct, indirect, and total effects.**

| Path | Direct effect | | Total indirect effect | | Total effect | |
|---|---|---|---|---|---|---|
| | Mean (SD) | P value | Mean (SD) | P value | Mean (SD) | P value |
| Age -->MH | 0.06 (0.05) | 0.26 | 0.11 (0.02) | <0.001 | 0.16 (0.06) | <0.001 |
| Age -->SI | −0.19 (0.04) | <0.001 | | | −0.19 (0.04) | <0.001 |
| Age-->BMI | | | −0.05 (0.02) | 0.01 | −0.05 (0.02) | 0.01 |
| BMI -->MH | −0.04 (0.05) | 0.43 | | | −0.04 (0.05) | 0.43 |
| CIAC -->ISC | −0.12 (0.06) | 0.05 | −0.09 (0.02) | <0.001 | −0.21 (0.06) | <0.001 |
| CIAC -->PCQ | −0.34 (0.05) | <0.001 | | | −0.34 (0.05) | <0.001 |
| CIAC-->SI | −0.09 (0.05) | 0.06 | −0.09 (0.03) | <0.001 | −0.18 (0.05) | <0.001 |
| CIAC-->BMI | | | −0.05 (0.02) | 0.01 | −0.05 (0.02) | 0.01 |
| CIAC-->MH | | | 0.10 (0.03) | <0.001 | 0.10 (0.03) | <0.001 |
| CIAC-->NLP | | | −0.08 (0.02) | <0.001 | −0.08 (0.02) | <0.001 |
| ISC-->NLP | 0.39 (0.05) | <0.001 | | | 0.39 (0.05) | <0.001 |
| ISC-->SI | 0.16 (0.06) | 0.01 | 0.18 (0.03) | <0.001 | 0.35 (0.06) | <0.001 |
| ISC-->BMI | | | 0.09 (0.02) | <0.001 | 0.09 (0.02) | <0.001 |
| ISC-->MH | | | −0.19 (0.03) | <0.001 | −0.19 (0.03) | <0.001 |
| NLP-->SI | 0.47 (0.05) | <0.001 | | | 0.47 (0.05) | <0.001 |
| NLP-->BMI | | | 0.12 (0.03) | <0.001 | 0.12 (0.03) | <0.001 |
| NLP-->MH | | | −0.26 (0.04) | <0.001 | −0.26 (0.04) | <0.001 |
| PCQ-->ISC | 0.26 (0.05) | <0.001 | | | 0.26 (0.05) | <0.001 |
| PCQ-->SI | 0.06 (0.04) | 0.17 | 0.09 (0.02) | <0.001 | 0.15 (0.04) | <0.001 |
| PCQ-->BMI | | | 0.04 (0.01) | 0.01 | 0.04 (0.01) | 0.01 |
| PCQ-->MH | | | −0.08 (0.03) | <0.001 | −0.08 (0.03) | <0.001 |
| PCQ-->NLP | | | 0.10 (0.02) | <0.001 | 0.10 (0.02) | <0.001 |
| SI-->BMI | 0.26 (0.06) | <0.001 | | | 0.26 (0.06) | <0.001 |
| SI-->MH | −0.54 (0.05) | <0.001 | −0.01 (0.01) | 0.44 | −0.55 (0.04) | <0.001 |
| TDX-->MH | −0.05 (0.05) | 0.24 | −0.07 (0.02) | <0.001 | −0.12 (0.05) | 0.01 |
| TDX-->SI | 0.13 (0.04) | <0.001 | | | 0.13 (0.04) | <0.001 |
| TDX-->BMI | | | 0.03 (0.01) | 0.01 | 0.03 (0.01) | 0.01 |
| **Quadratic effects (QE)** | | | | | | |
| Age-->MH | −0.13 (0.04) | <0.001 | | | −0.13 (0.04) | <0.001 |

Social isolation was one of the strongest predictors of mental health, with a direct negative effect of β = −0.54 (p < 0.001) and a total effect of β = −0.55 (p < 0.001), indicating a strong link between social disconnection and psychological distress. Time since cancer diagnosis (TDX) was not directly associated with mental health (β = −0.05, p = 0.24) but had a significant total effect (β = −0.12, p = 0.01), likely mediated by other constructs. The quadratic effect of age on mental health was negative (β = −0.13, p < 0.001), suggesting a non-linear relationship where mental health declines more steeply with age beyond a certain threshold.

Fig 3 illustrates the structural model summarizing all the paths influencing social isolation and mental health.

## Multigroup analysis

The multigroup analysis, as shown in Table 6, indicates differences in the significance of predictors between urban and rural populations.

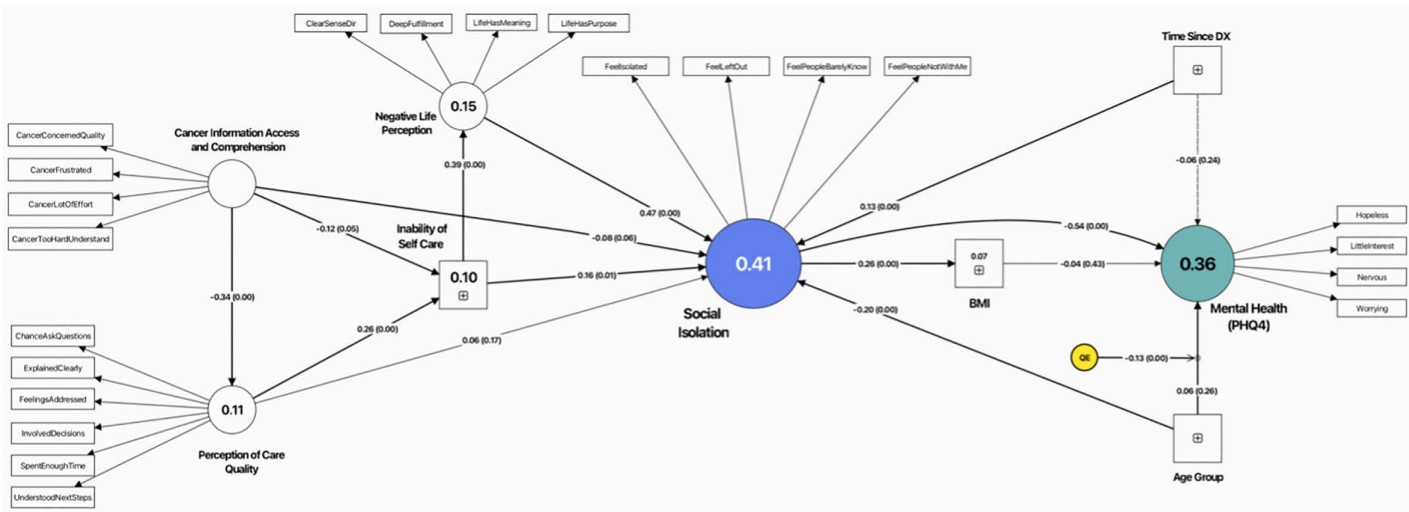

**Fig 3. Structural equation model depicting the relationships between cancer information access and comprehension, perception of care quality, social isolation, negative life perception, inability of self-care, BMI, time since diagnosis, age, and mental health (PHQ-4).** Standardized path coefficients (β) with significance levels are shown along the arrows, and R² values indicate the explained variance for social isolation and mental health. QE represents quadrative effect.

Age significantly predicted mental health (MH) in both groups, but the effect was slightly stronger in the rural group (β = 0.21, p = 0.02) compared to the urban group (β = 0.16, p = 0.02). Age also negatively predicted social isolation (SI) in both groups, but again, the effect was more pronounced in the rural group (β = −0.21, p = 0.03) compared to the urban group (β = −0.19, p < 0.001). However, the relationship between age and BMI was significant only in the urban group (β = −0.04, p = 0.03) but not in the rural group (β = −0.09, p = 0.11).

Cancer Information Access and Comprehension (CIAC) significantly predicted Perception of Care Quality (PCQ) in both groups, but its effect was stronger in the rural group (β = −0.39, p = 0.01) than in the urban group (β = −0.34, p < 0.001). However, CIAC significantly predicted mental health (β = 0.10, p < 0.001) only in the urban group, while this relationship was non-significant in the rural group (β = 0.15, p = 0.19). CIAC also significantly predicted social isolation (β = −0.18, p < 0.001) in the urban group but not in the rural group (β = −0.24, p = 0.14).

Inability of Self-Care (ISC) significantly influenced negative life perception (NLP) in both groups, but its effect on social isolation was significant only in the urban group (β = 0.40, p < 0.001) and non-significant in the rural group (β = 0.14, p = 0.48). Similarly, ISC significantly predicted mental health (β = −0.22, p < 0.001) in the urban group, but not in the rural group (β = −0.07, p = 0.45).

Social isolation emerged as a critical predictor of mental health in both groups, with a strong negative effect in the urban group (β = −0.55, p < 0.001) and an even stronger effect in the rural group (β = −0.59, p < 0.001). However, its effect on BMI was more pronounced in the rural group (β = 0.39, p = 0.01) than in the urban group (β = 0.20, p < 0.001).

## Discussion

This study highlights the factors that contribute to perceived social isolation (SI) and how they may affect mental health, particularly depression, among cancer survivors. Our findings point to the need for targeted interventions, such as strengthening social support and improving access to clear and useful cancer-related information, to help improve overall well-being in this population. Specifically, the study identified cancer information access and comprehension (CIAC), inability of self-care (ISC), time since diagnosis (TDx), negative life perception (NLP), and perception of care quality

**Table 6. Standardized Total effects. (Urban vs Rural).**

| Path | Urban Group (N = 783) | | Rural Group (N = 143) | |
|---|---|---|---|---|
| | Mean (SD) | P value | Mean (SD) | P value |
| Age -->MH | 0.16 (0.07) | 0.02 | 0.21 (0.10) | 0.02 |
| Age -->SI | −0.19 (0.05) | <0.001 | −0.21 (0.10) | 0.03 |
| Age-->BMI | −0.04 (0.02) | 0.03 | −0.09 (0.06) | 0.11 |
| BMI -->MH | −0.08 (0.06) | 0.17 | 0.06 (0.12) | 0.62 |
| CIAC -->ISC | −0.23 (0.06) | <0.001 | −0.07 (0.18) | 0.76 |
| CIAC -->PCQ | −0.34 (0.05) | <0.001 | −0.39 (0.13) | 0.01 |
| CIAC-->SI | −0.18 (0.06) | <0.001 | −0.24 (0.18) | 0.14 |
| CIAC-->BMI | −0.04 (0.02) | 0.02 | −0.10 (0.09) | 0.25 |
| CIAC-->MH | 0.10 (0.03) | <0.001 | 0.154 (0.12) | 0.19 |
| CIAC-->NLP | −0.09 (0.03) | <0.001 | −0.03 (0.08) | 0.75 |
| ISC-->NLP | 0.38 (0.05) | <0.001 | 0.43 (0.10) | <0.001 |
| ISC-->SI | 0.40 (0.05) | <0.001 | 0.14 (0.14) | 0.48 |
| ISC-->BMI | 0.08 (0.03) | <0.001 | 0.05 (0.05) | 0.45 |
| ISC-->MH | −0.22 (0.03) | <0.001 | −0.07 (0.08) | 0.45 |
| NLP-->SI | 0.45 (0.06) | <0.001 | 0.56 (0.11) | <0.001 |
| NLP-->BMI | 0.09 (0.02) | <0.001 | 0.21 (0.08) | <0.001 |
| NLP-->MH | −0.25 (0.04) | <0.001 | −0.33 (0.09) | <0.001 |
| PCQ-->ISC | 0.26 (0.05) | <0.001 | 0.31 (0.11) | 0.01 |
| PCQ-->SI | 0.17 (0.05) | <0.001 | 0.12 (0.08) | 0.15 |
| PCQ-->BMI | 0.03 (0.01) | 0.02 | 0.05 (0.04) | 0.22 |
| PCQ-->MH | −0.09 (0.03) | <0.001 | −0.07 (0.05) | 0.19 |
| PCQ-->NLP | 0.10 (0.02) | <0.001 | 0.13 (0.06) | 0.02 |
| SI-->BMI | 0.20 (0.06) | <0.001 | 0.39 (0.15) | 0.01 |
| SI-->MH | −0.55 (0.04) | <0.001 | −0.59 (0.11) | <0.001 |
| TDx-->MH | −0.11 (0.06) | 0.05 | −0.18 (0.07) | 0.01 |
| TDx-->SI | 0.11 (0.04) | 0.01 | 0.17 (0.08) | 0.01 |
| TDx-->BMI | 0.02 (0.01) | 0.05 | 0.06 (0.04) | 0.04 |
| **Quadratic effects (QE)** | | | | |
| Age-->MH | −0.11 (0.05) | 0.01 | −0.14 (0.10) | 0.11 |

(PCQ) as key factors that lead to social isolation. These findings reinforce the critical role of social isolation by demonstrating its potential impact on mental health outcomes. Note, readers should interpret the findings and any corresponding recommendations with caution, as they are derived from cross-sectional data. This secondary data limits our ability to establish temporal ordering or infer causality with certainty. The observed associations reflect correlations at a single point in time and may be influenced by unmeasured factors or reverse causation. Therefore, these results should be viewed as hypothesis-generating rather than determinative, and future longitudinal or experimental studies are needed to confirm the causal mechanisms implied by our model.

## Access to cancer information and comprehension

CIAC had significant negative total effects on SI suggesting that when patients have better access to clear, relevant, and understandable cancer-related information, they are more likely to feel connected, informed, and emotionally supported. This information may have been directly relevant to social support, such as those provided by the American Cancer

Society [16]. This finding aligns with growing evidence that health literacy and effective communication play crucial roles in shaping cancer patients' psychological well-being [17]. For example, patients who understand their diagnosis, treatment options, and expected side effects are more likely to engage in informed decision-making, maintain a sense of control, and feel more connected to their care team. This can potentially reduce feelings of uncertainty, fear, and helplessness, common drivers of social withdrawal and emotional distress. Enhanced CIAC may also reduce perceived stigma or isolation by encouraging open conversations with family members and peers, which is particularly important in cultures or communities where discussing cancer may be taboo [18].

### Inability of self-care

ISC showed a strong association with SI, suggesting that when cancer survivors face challenges in managing their daily health needs, they are more likely to feel disconnected and unsupported. This may be especially true in the post-treatment phase, when formal medical oversight decreases and survivors often find themselves without the structured support they had during active treatment. For example, while patients undergoing chemotherapy may receive regular contact from healthcare providers and emotional support from family and friends, this support may diminish once treatment ends, leaving survivors to navigate lingering symptoms, fatigue, and emotional distress on their own. TDx further supports the role of ISC in contributing to social isolation, as the study found that SI tends to increase as more time passes after a cancer diagnosis. This cumulative effect likely reflects the progressive nature of self-care challenges and the gradual decline in external support, making it harder for survivors to stay socially active and emotionally connected. This gap in continuity of care could make it harder to maintain independence and social engagement, exacerbating feelings of isolation. These findings highlight the importance of integrating long-term self-management support into survivorship care plans, equipping patients with strategies to manage symptoms and maintain social participation, echoing prior research advocating for ongoing, tailored self-management education as a critical component of survivorship care to promote both psychological and physical well-being [19,20].

### Negative perception of life

The effect of NLP on SI highlights the critical role of survivors' outlook on life in shaping their social experiences. When cancer survivors perceive their lives negatively, due to ongoing physical limitations, emotional distress, financial strain, or a diminished sense of purpose, they may be less likely to seek out or maintain social connections. This negative worldview can potentially lead to withdrawal from social activities, reluctance to engage with support networks, or feelings of being misunderstood, all of which contribute to a growing sense of isolation. For example, a survivor who feels hopeless about their future or views their quality of life as poor may avoid interactions with family and friends, believing they are a burden or that others cannot relate to their experience. Such perception-driven disconnection creates a reinforcing cycle, where isolation further deepens negative life views and vice versa. These findings emphasize the importance of addressing survivors' cognitive and emotional appraisals of their life circumstances as part of psychosocial care. Notably, the pathway from NLP to SI, and subsequently from SI to poorer mental health, suggests a cascading effect that amplifies psychological vulnerability. However, this relationship may also operate in the opposite direction: mental health challenges such as depression and anxiety may contribute to increased SI, which in turn can intensify NLP. This potential bidirectionality across constructs surfaces the complex nature of such interrelationships. Readers should keep this in mind when interpreting our findings, as the cross-sectional design limits causal inference.

### Perception of care quality

The relationship between PCQ and SI underscores the broader influence of healthcare experiences on cancer survivors' social and emotional well-being. When individuals perceive the care, they receive as inadequate, impersonal, or poorly coordinated, it can erode their trust in the healthcare system and contribute to feelings of neglect or abandonment. These

negative perceptions may reduce motivation to engage with care teams, attend follow-up appointments, or participate in survivorship programs, opportunities that often serve as important sources of social connection and emotional support. For example, a survivor who feels dismissed by their providers or perceives a lack of empathy during treatment may internalize these experiences as a reflection of their value or worth, leading to withdrawal from both medical and social environments. This detachment can amplify feelings of isolation, particularly if survivors do not feel heard or supported in managing long-term side effects or emotional challenges. These findings point to the need for patient-centered care models that prioritize clear communication, emotional responsiveness, and continuity of care throughout the survivorship trajectory. By fostering a positive care experience, healthcare providers may help reinforce a sense of being cared for and connected factors that may buffer against SI.

### Social isolation

SI emerged as one of the strongest predictors of MH in this study, reinforcing a substantial body of evidence that highlights the profound psychological toll of social disconnection among cancer survivors. This finding resonates with literature emphasizing the detrimental effects of social isolation on psychological well-being, particularly among cancer survivors who often experience loneliness due to their condition [19,21,22]. Sustained periods of loneliness since diagnosis may exacerbate symptoms of depression and anxiety, limit help-seeking behaviors, and erode quality of life [23]. For instance, a survivor who feels isolated may lack the emotional encouragement needed to adhere to treatment recommendations or to engage in health-promoting behaviors, thereby further compromising their mental well-being. This finding underscores the critical importance of understanding the antecedents of SI which were identified in this study. By recognizing and addressing the upstream factors that contribute to isolation, healthcare providers and policymakers can design more effective, tailored interventions to prevent its onset.

Importantly, while our findings emphasize the predictive role of social isolation on mental health, it is important to acknowledge the likely bidirectional nature of such relationship. SI may contribute to the onset or worsening of mental health conditions such as depression and anxiety; however, these same mental health conditions can also exacerbate social disconnection. While our model presents unidirectional paths due to the cross-sectional nature of the data, the bidirectionality should be considered when drawing inferences or designing interventions based on our results.

### Antecedents of isolation and depression in urban and rural setting

The comparison of predictors across urban and rural groups surfaces differences in how various factors influence social isolation and mental health, suggesting that the context of where a cancer survivor lives may shape their experience in unique ways. In both settings, social isolation strongly predicted poorer mental health, with slightly stronger effects in the rural group compared to the urban group, reinforcing the universal importance of addressing isolation in survivorship care. However, the significance and magnitude of other predictors varied notably between the two groups.

CIAC significantly impacted SI and MH in the urban group but did not show a significant effect in the rural group. This discrepancy may reflect underlying disparities in access to digital resources, differences in health literacy, and varying capacities to navigate complex healthcare systems [24,25]. In urban areas, cancer survivors are more likely to have access to broadband internet, patient portals, health apps, and informational materials. They may also have more frequent interactions with specialists, educational workshops, or support groups that help them understand their diagnosis and treatment plans. These resources can foster a sense of empowerment and connection, thereby reducing isolation and emotional distress. In contrast, survivors in rural settings may face barriers to obtaining and comprehending reliable cancer-related information [26,27]. Limited internet access, lower availability of cancer education programs, and fewer healthcare providers contribute to a gap in informational support. Furthermore, even when information is available, it may not be presented in ways that are understandable to individuals with lower literacy levels. As a result, survivors may struggle to interpret medical information, make informed decisions, or feel confident in their care, which can lead to increased

feelings of vulnerability and disconnection. This lack of comprehension may weaken the potential protective effects of information access on mental health and social isolation in rural populations.

Similarly, ISC had a strong and significant impact on SI in the urban group but was not significant in the rural group, suggesting that challenges in managing daily health needs may carry different social consequences depending on the context [28]. In urban settings, self-sufficiency and independence are often emphasized, and survivors may be expected to manage their appointments, medications, and day-to-day functioning with minimal assistance. When individuals in these environments struggle with self-care, due to fatigue, cognitive impairments, or physical limitations, they may find it harder to participate in social activities, maintain employment, or keep up with fast-paced urban life. These limitations can lead to social withdrawal, a shrinking support network, and ultimately, heightened isolation. In contrast, rural communities may be characterized by tighter-knit social networks, where interdependence and community-based support are more culturally accepted and expected [29]. Survivors who struggle with self-care in rural areas may receive informal help from family, neighbors, or local community members, which could buffer the impact of their limitations on social isolation [30]. For example, a rural cancer survivor may rely on a neighbor for transportation or on extended family for household tasks, forms of support that may be less readily available or offered in urban environments. Moreover, the pace and structure of life in rural areas may be more accommodating to those with limited physical ability, reducing the social consequences of self-care challenges.

The effect of TDx on MH was significant in the rural group but not in the urban group, suggesting that the emotional toll of survivorship may intensify over time in settings with fewer formal resources [31–34]. While rural survivors may initially benefit from strong family and community support, this support often diminishes as time passes, leaving individuals to cope with lingering symptoms, fear of recurrence, and emotional fatigue on their own. Unlike urban areas where ongoing survivorship care and mental health services are more accessible [35]. These findings highlight the potential need for sustained mental health support in rural areas, such as telehealth counseling, follow-up care, and community-based interventions tailored to the unique challenges of rural cancer survivors [36,37].

Lastly, the effects of PCQ on SI and MH being significant only in the urban group may reflect structural and systemic differences in how care is delivered across settings. One speculative explanation is that in urban healthcare systems, high patient volumes, time-limited consultations, and administrative demands may hinder the development of strong patient–provider relationships [38]. Survivors who perceive their care as impersonal or fragmented may feel emotionally unsupported, which could contribute to both social withdrawal and psychological distress [39]. Additionally, higher healthcare costs and greater system complexity in urban areas, combined with variable health literacy, may further distance patients from engaging meaningfully with their care [40,41]. In contrast, rural survivors, despite having fewer specialized resources, may benefit from more continuous and personalized care within smaller systems, potentially buffering the impact of lower perceived care quality. While these interpretations remain speculative, they point to important avenues for future research to better understand how healthcare system dynamics influence patient perceptions and downstream psychosocial outcomes in different geographic contexts.

## Future work and limitations

Our study underscores the need for context-specific understanding and intervention to support cancer survivors. For instance, we found varied antecedents of social isolation in urban and rural populations. This suggests that in urban settings, where healthcare systems may be more fragmented and fast-paced, patients with lower self-care capacity and limited access to comprehensible cancer information may become more socially disconnected. In contrast, rural communities may face broader structural challenges, such as geographic isolation, limited healthcare infrastructure, and transportation barriers, that influence social isolation in different ways, not fully captured in the current model. Moreover, cultural norms around mental health stigma and community engagement likely differ between settings, further shaping experiences of isolation and emotional distress. Future research should analyze these urban–rural disparities

to uncover the mechanisms driving social isolation and depression in different settings. Such work is essential for developing tailored interventions and policies that address the unique social, cultural, and structural contexts of both urban and rural cancer survivors.

Future studies should also incorporate the accumulation and progression of patient stress over time, which may provide a more comprehensive understanding of how chronic stressors interact with personal and environmental resources to affect health outcomes. For example, repeated exposures to financial hardship, inadequate informational support, or persistent self-care difficulties may have compounding effects on social isolation and psychological well-being that unfold gradually. Integrating cumulative stress trajectories into future models could also help explain delayed or nonlinear effects that are not visible in cross-sectional snapshots.

In addition, while structural equation modeling allows for the examination of theoretically informed pathways, it does not establish temporality or causality. For example, although our model suggests that social isolation leads to poorer mental health and diminished perceptions of care quality, it is equally plausible that individuals with preexisting mental health challenges or dissatisfaction with care may be more likely to withdraw socially. Future longitudinal studies are essential to validate the proposed relationships, clarify the temporal ordering of variables, and better capture dynamic changes in psychosocial stressors and health outcomes over time.

Several other limitations should be considered when interpreting the findings of this study. First, although we focused on cancer survivors, the inclusion criteria were broad and included individuals with a history of skin cancer. Skin cancer, particularly non-melanoma types, may differ in its psychological and treatment burden compared to other cancers, potentially influencing the generalizability of the results to survivors of more intensive or life-threatening cancer types. Survivors with multiple cancers may have different psychosocial experiences and care needs compared to those with a single diagnosis, and this heterogeneity could not be accounted for in our analysis.

Second, the study relied on secondary data from the Health Information National Trends Survey (HINTS), which limited our ability to include certain theoretically relevant variables. Key constructs such as comorbidities, symptom severity, and detailed treatment history were not consistently available or measurable using the items provided. As a result, our conceptual framework was necessarily shaped by the variables that were accessible in the dataset. While our model demonstrated good fit and explanatory power, it may not capture all relevant mediators or moderators, especially cultural, geographic, or system-level factors that shape survivorship experiences. Future research should aim to validate this framework with expanded measures that include structural determinants of health and longitudinal change. In addition to the measurement constraints, several critically important variables were absent from HINTS, further limiting the interpretability of our findings. Notably, the dataset does not capture cancer stage, treatment modality, or the strength of patients' social and support networks, factors known to strongly influence psychological distress, coping behaviors, and health communication patterns. Other missing factors that warrant consideration in future studies include cancer recurrence status, treatment side-effect burden, functional limitations, insurance adequacy, transportation barriers, and neighborhood-level environmental exposures. These factors can shape perceived distress, help-seeking behavior, and disparities in health outcomes but could not be evaluated within the constraints of the HINTS dataset. The absence of these variables constrains our ability to contextualize the observed associations and may mask important subgroup differences or pathways. Future research should aim to validate this framework with expanded measures that include clinically, psychosocially, and structurally salient variables, ideally using datasets that incorporate these determinants and allow for longitudinal assessment.

Third, the use of self-reported data introduces the potential for recall and social desirability biases, particularly in the assessment of sensitive variables such as mental health status and perceived quality of care.

Despite these limitations, the study provides important insights into the psychosocial pathways affecting mental health among cancer survivors and highlights modifiable areas for intervention, particularly in the domains of informational support and social connectedness.

## Author contributions

**Conceptualization:** Avishek Choudhury, Christopher W. Wheldon.

**Data curation:** Avishek Choudhury, Christopher W. Wheldon, Safa Elkefi.

**Formal analysis:** Avishek Choudhury, Christopher W. Wheldon.

**Investigation:** Avishek Choudhury.

**Methodology:** Avishek Choudhury, Christopher W. Wheldon.

**Project administration:** Avishek Choudhury.

**Supervision:** Avishek Choudhury.

**Validation:** Avishek Choudhury, Christopher W. Wheldon, Safa Elkefi.

**Visualization:** Avishek Choudhury.

**Writing – original draft:** Avishek Choudhury, Christopher W. Wheldon, Safa Elkefi, Priyanka Dixit.

**Writing – review & editing:** Avishek Choudhury, Christopher W. Wheldon, Safa Elkefi, Priyanka Dixit.

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
