## [Decision Letter · Decision Letter 0]

10 Jul 2025

Dear Dr. Choudhury,

Thank you for submitting your manuscript to PLOS ONE. After careful consideration, we feel that it has merit but does not fully meet PLOS ONE’s publication criteria as it currently stands. Therefore, we invite you to submit a revised version of the manuscript that addresses the points raised during the review process.

**
*Thank you for your submission. Please provide a point-by-point response to all the comments.*
**

We look forward to receiving your revised manuscript.

Kind regards,

Mohammad Mofatteh, PhD, MPH, MSc, PGCert, BSc (Hons), MB BCh (c)

Academic Editor

PLOS ONE

Journal Requirements:

3. We note you have included a table to which you do not refer in the text of your manuscript. Please ensure that you refer to Table 5 in your text; if accepted, production will need this reference to link the reader to the Table.

Additional Editor Comments:

Thank you for your submission. Please provide a point-by-point response to all the comments.

Reviewers' comments:

Reviewer's Responses to Questions

**Comments to the Author**

1. Is the manuscript technically sound, and do the data support the conclusions?

Reviewer #1: Yes

Reviewer #2: Yes

Reviewer #3: Yes

2. Has the statistical analysis been performed appropriately and rigorously?

Reviewer #1: Yes

Reviewer #2: Yes

Reviewer #3: Yes

3. Have the authors made all data underlying the findings in their manuscript fully available?

Reviewer #1: Yes

Reviewer #2: Yes

Reviewer #3: Yes

4. Is the manuscript presented in an intelligible fashion and written in standard English?

Reviewer #1: Yes

Reviewer #2: Yes

Reviewer #3: Yes

Reviewer #1: Thank you for the chance to review the paper. The only noticeable flaw is the limited number of patients in the urban setting. Other limitations have been stated in the limitations section. This paper can inspire many future researchers to investagate the topic further.

Reviewer #2: Thank you for the opportunity to review this manuscript, which explore antecedents of social isolation and their impact on mental health among cancer survivors. This is a strong and well-executed paper. I recommend only minor revisions to further strengthen its conceptual clarity and interpretive depth.

Bidirectionality of SIL and Mental Health:

One central question concerns the relationship between social isolation and loneliness (SIL) and mental health. A complicating factor is the likely bidirectional relationship between these constructs. SIL may contribute to the development of mental health challenges, but it may also arise as a consequence of such conditions—for example, anhedonia and social withdrawal are hallmark symptoms of depression. I encourage the authors to briefly address this bidirectionality in the discussion section, even if only to note the limitations of causal inference in the current design.

A similar consideration applies to variables related to negative perceptions of life, which may serve both as predisposing risk factors and as symptoms of depression Acknowledging this complexity would strengthen the interpretive nuance of the paper.

Use of Theory and Modeling Approach:

The application of theory and statistical modeling is logical, rigorous, and clearly explained. The incorporation of the Stress Process Model is a strength. As described, this model highlights the effects of cumulative stress over time. However, it was unclear to what extent the cumulative nature of stress exposure was captured in the analytic model. If not explicitly modeled, the authors may consider discussing how the model could inform future work or how the cumulative nature of stress might shape the interpretation of their findings.

Reviewer #3: .

The study addresses a relevant topic, focusing on a determinant of mental and physical health: social isolation in cancer survivors. It uses a nationally representative database (HINTS 6), allowing important generalizations to the US population. PLS-SEM modeling is suitable for examining complex and mediating relationships in contexts with multiple latent variables. The division between urban and rural areas introduces an important contextual dimension to the analysis, thereby expanding the understanding of geographic disparities in cancer care.

Overall, this is a rigorous and pertinent study that contributes to the understanding of the factors associated with social isolation and its relationship with mental health among cancer survivors. However, the manuscript could benefit from some improvements, mainly in the detailing of certain information (such as the sample), in the discussion of the implications of the limitations in the results, and in including a section on the practical implications of the obtained results.

1. A more detailed description of the sample is missing, mainly in terms of sociodemographic and clinical variables. Although the study includes basic information, the authors should include a more complete table of sociodemographic and clinical characteristics (such as age, ethnicity/race, income, type of cancer, etc.). It may make sense to separate the descriptive statistics between urban and rural areas (since this is a focus of the study). This more detailed description provides greater richness to the study and contextualization.

2. Since this is a cross-sectional study, limiting causal inferences about the relationships proposed in the theoretical model, it would be pertinent to include: a more explicit discussion about the limitations of causality in cross-sectional studies; suggest future longitudinal studies to validate the proposed relationships; reflect on possible reversibility bias in the observed relationships (e.g., isolation → poorer perceived quality of care).

3. Explicitly discuss which important variables were missing in the HINTS data and how this may affect the results

4. All variables were self-reported; therefore, other types of bias, such as recall bias and social desirability, should be discussed.

5. Provide better support for interpretations based on existing literature on the differences between rural and urban health systems or contexts.

6. An essential component that is missing from the manuscript is directly suggesting practical implications based on the results obtained (e.g., public policies, contextual interventions, etc.)

If the suggested improvements are implemented, the manuscript can be accepted into PLOS ONE and make a significant contribution to the literature and scientific research in oncology, mental health, and public health.

**Do you want your identity to be public for this peer review?** For information about this choice, including consent withdrawal, please see our Privacy Policy

Reviewer #1: No

Reviewer #2: No

Reviewer #3: No

---

## [Author Response · Author response to Decision Letter 1]

23 Jul 2025

Reviewer #1:

Thank you for the chance to review the paper. The only noticeable flaw is the limited number of patients in the urban setting. Other limitations have been stated in the limitations section. This paper can inspire many future researchers to investagate the topic further.

Response: Thank you very much. We have acknowledged it as a limitation.

Reviewer #2:

Thank you for the opportunity to review this manuscript, which explore antecedents of social isolation and their impact on mental health among cancer survivors. This is a strong and well-executed paper. I recommend only minor revisions to further strengthen its conceptual clarity and interpretive depth.

One central question concerns the relationship between social isolation and loneliness (SIL) and mental health. A complicating factor is the likely bidirectional relationship between these constructs. SIL may contribute to the development of mental health challenges, but it may also arise as a consequence of such conditions—for example, anhedonia and social withdrawal are hallmark symptoms of depression. I encourage the authors to briefly address this bidirectionality in the discussion section, even if only to note the limitations of causal inference in the current design.

Response: Thank you for this important observation. We agree that the relationship between social isolation and loneliness (SIL) and mental health is likely bidirectional, with each influencing the other over time. In the revised Discussion section, we have acknowledged this complexity by noting that while SIL can increase vulnerability to mental health challenges such as depression and anxiety, these same conditions can, in turn, exacerbate SIL through mechanisms like social withdrawal, reduced motivation, and anhedonia. We have also highlighted the need for future longitudinal or causal modeling studies to disentangle these temporal dynamics and better understand how interventions might interrupt this feedback loop.

A similar consideration applies to variables related to negative perceptions of life, which may serve both as predisposing risk factors and as symptoms of depression Acknowledging this complexity would strengthen the interpretive nuance of the paper.

Response: We have acknowledged this as well.

The application of theory and statistical modeling is logical, rigorous, and clearly explained. The incorporation of the Stress Process Model is a strength. As described, this model highlights the effects of cumulative stress over time. However, it was unclear to what extent the cumulative nature of stress exposure was captured in the analytic model. If not explicitly modeled, the authors may consider discussing how the model could inform future work or how the cumulative nature of stress might shape the interpretation of their findings.

Response: You are correct in noting that the cumulative nature of stress exposure, as emphasized in the SPM, was not directly captured in our statistical model due to the cross-sectional nature of the data. Our use of the SPM was primarily to provide a theoretical foundation for understanding the interrelated psychosocial and health constructs examined in the study. We agree that modeling cumulative stress over time would offer important additional insights, and we have acknowledged this limitation in the manuscript. We have also noted in the future research section that longitudinal designs are needed to explore how stress accumulates and evolves, and to examine its dynamic effects on outcomes such as social isolation, mental health, and life perception. We have revised the manuscript to further clarify this point and ensure readers are aware of the theoretical versus empirical application of the model in this context.

Reviewer #3:

The study addresses a relevant topic, focusing on a determinant of mental and physical health: social isolation in cancer survivors. It uses a nationally representative database (HINTS 6), allowing important generalizations to the US population. PLS-SEM modeling is suitable for examining complex and mediating relationships in contexts with multiple latent variables. The division between urban and rural areas introduces an important contextual dimension to the analysis, thereby expanding the understanding of geographic disparities in cancer care.

Overall, this is a rigorous and pertinent study that contributes to the understanding of the factors associated with social isolation and its relationship with mental health among cancer survivors. However, the manuscript could benefit from some improvements, mainly in the detailing of certain information (such as the sample), in the discussion of the implications of the limitations in the results, and in including a section on the practical implications of the obtained results.

1. A more detailed description of the sample is missing, mainly in terms of sociodemographic and clinical variables. Although the study includes basic information, the authors should include a more complete table of sociodemographic and clinical characteristics (such as age, ethnicity/race, income, type of cancer, etc.). It may make sense to separate the descriptive statistics between urban and rural areas (since this is a focus of the study). This more detailed description provides greater richness to the study and contextualization.

Response: We agree. We have added a new table with participant characteristics.

2. Since this is a cross-sectional study, limiting causal inferences about the relationships proposed in the theoretical model, it would be pertinent to include: a more explicit discussion about the limitations of causality in cross-sectional studies; suggest future longitudinal studies to validate the proposed relationships; reflect on possible reversibility bias in the observed relationships (e.g., isolation → poorer perceived quality of care).

Response: We agree and have revised the future work section as per your suggestion. We did previously acknowledge the causality limitation but in this revision, we have added the potential of bi-directionality between variable as you and other reviewers identified.

Just to clarify, while developing the model, we did test all possible permutation and combinations of path-directions and finalized this model due to it best performance and better alignment with some existing theories.

3. Explicitly discuss which important variables were missing in the HINTS data and how this may affect the results

Response: We mentioned the missing data handling in the method section. “Missing values were handled using the pairwise deletion method, allowing the retention of as much data as possible while excluding only the missing values for specific analyses.”

PLS-SEM is designed to handle smaller data size and pairwise deletion method minimizes skewness in the results. To further safeguard the model, we used 10,000 bootstrapping iterations. Stating the same in the discussion will be redundant.

4. All variables were self-reported; therefore, other types of bias, such as recall bias and social desirability, should be discussed.

Response: We have this in the limitation section.

5. Provide better support for interpretations based on existing literature on the differences between rural and urban health systems or contexts.

Response: This is beyond the scope. HINTS don’t provide us with health systems settings. We cannot control for it. A respondent can be in a rural area but is not necessarily suffering poverty. (We don’t have this information). Additionally, our model does not account for any health system components or SEIPS model. So, comparing based on health system will only be a stretched inference not based on the direct findings. However, we did compare those for which we noted significant differences based in page 28 – 30.

6. An essential component that is missing from the manuscript is directly suggesting practical implications based on the results obtained (e.g., public policies, contextual interventions, etc.)

Response: We cannot suggest these based on non-causal findings. It will be scientifically incorrect.

---

## [Decision Letter · Decision Letter 1]

14 Oct 2025

Dear Dr. Choudhury,

Thank you for submitting your manuscript to PLOS ONE. After careful consideration, we feel that it has merit but does not fully meet PLOS ONE’s publication criteria as it currently stands. Therefore, we invite you to submit a revised version of the manuscript that addresses the points raised during the review process.

**The manuscript has been evaluated by three reviewers, and their comments are available below.**

**Although reviewers 1 and 2 are satisfied with the revised manuscript, reviewer 3 has some requests.**

**Could you please revise the manuscript to carefully address the concerns raised?**

We look forward to receiving your revised manuscript.

Kind regards,

Steve Zimmerman, PhD

Senior Editor, PLOS One

**Journal Requirements:**

Reviewers' comments:

Reviewer's Responses to Questions

**Comments to the Author**

Reviewer #1: All comments have been addressed

Reviewer #2: All comments have been addressed

Reviewer #3: All comments have been addressed

2. Is the manuscript technically sound, and do the data support the conclusions?

Reviewer #1: Yes

Reviewer #2: Yes

Reviewer #3: Yes

3. Has the statistical analysis been performed appropriately and rigorously?

Reviewer #1: Yes

Reviewer #2: Yes

Reviewer #3: Yes

4. Have the authors made all data underlying the findings in their manuscript fully available?

Reviewer #1: (No Response)

Reviewer #2: Yes

Reviewer #3: Yes

5. Is the manuscript presented in an intelligible fashion and written in standard English?

Reviewer #1: Yes

Reviewer #2: Yes

Reviewer #3: Yes

**Reviewer #1:** (No Response)

**Reviewer #2:** The authors have sufficiently address my feedback and I am happy to recommend this paper for publication.

**Reviewer #3:** The manuscript has been significantly improved, but still requires some minor adjustments that will further strengthen the study's contribution to the literature on cancer survivor health.

The authors considered and responded to all comments and suggestions. A new table with participant characteristics was added, increasing the transparency and contextualization of the study.

The discussion of the limitations of cross-sectional studies was adequately addressed. The inclusion of the potential bidirectionality between variables (e.g., social isolation ↔ mental health) and the suggestion of future longitudinal studies are strengths of the new version of the article.

Regarding missing variables in HINTS, the authors refer to the treatment of missing data. However, what was suggested was a critical discussion of the important variables missing from HINTS and how this may limit the interpretation of the results (examples of critical missing variables include cancer type, treatment type, access to mental health services, and support network). We suggest that the authors engage in this critical reflection.

Regarding the literature on rural and urban contexts, the answer that this is "out of scope" is insufficient. Statistical control for health system variables was not requested; however, theoretical foundations from the existing literature were used to interpret the differences observed between rural and urban areas. It is suggested that studies that discuss access barriers, community cohesion, or disparities in post-cancer mental health be cited to better support interpretations, even if indirectly.

Regarding practical implications, caution must be exercised and causal prescriptions avoided. However, even in exploratory studies, it is possible and desirable to suggest practical implications and plausible interventions based on the results, always with some caution. For example, telemedicine programs, virtual support groups, or health literacy initiatives could be mentioned as potential strategies, especially in rural communities. We recommend including an "Implications for Practice" subsection with possible, data-based recommendations.

**Do you want your identity to be public for this peer review?** For information about this choice, including consent withdrawal, please see our Privacy Policy

Reviewer #1: No

Reviewer #2: No

Reviewer #3: No

---

## [Author Response · Author response to Decision Letter 2]

25 Nov 2025

Reviewer #1: (No Response)

Reviewer #2: The authors have sufficiently address my feedback and I am happy to recommend this paper for publication.

--Response: Thank you

Reviewer #3: The manuscript has been significantly improved, but still requires some minor adjustments that will further strengthen the study's contribution to the literature on cancer survivor health. The authors considered and responded to all comments and suggestions. A new table with participant characteristics was added, increasing the transparency and contextualization of the study. The discussion of the limitations of cross-sectional studies was adequately addressed. The inclusion of the potential bidirectionality between variables (e.g., social isolation ↔ mental health) and the suggestion of future longitudinal studies are strengths of the new version of the article.

--Response: Thank you for the acknowledgement

Regarding missing variables in HINTS, the authors refer to the treatment of missing data. However, what was suggested was a critical discussion of the important variables missing from HINTS and how this may limit the interpretation of the results (examples of critical missing variables include cancer type, treatment type, access to mental health services, and support network). We suggest that the authors engage in this critical reflection.

--Response: We have now added a critical discussion of the important missing variables from HINTS in the discussion section of the manuscript.

“In addition to the measurement constraints, several critically important variables were absent from HINTS, further limiting the interpretability of our findings. Notably, the dataset does not capture cancer stage, treatment modality, or the strength of patients’ social and support networks, factors known to strongly influence psychological distress, coping behaviors, and health communication patterns. Other missing factors that warrant consideration in future studies include cancer recurrence status, treatment side-effect burden, functional limitations, insurance adequacy, transportation barriers, and neighborhood-level environmental exposures. These factors can shape perceived distress, help-seeking behavior, and disparities in health outcomes but could not be evaluated within the constraints of the HINTS dataset. The absence of these variables constrains our ability to contextualize the observed associations and may mask important subgroup differences or pathways. Future research should aim to validate this framework with expanded measures that include clinically, psychosocially, and structurally salient variables, ideally using datasets that incorporate these determinants and allow for longitudinal assessment.”

Regarding the literature on rural and urban contexts, the answer that this is "out of scope" is insufficient. Statistical control for health system variables was not requested; however, theoretical foundations from the existing literature were used to interpret the differences observed between rural and urban areas. It is suggested that studies that discuss access barriers, community cohesion, or disparities in post-cancer mental health be cited to better support interpretations, even if indirectly.

--Response: We have added 8 new references. Please see ref# 29,30,35,36,37,38,40, and 41.

Regarding practical implications, caution must be exercised and causal prescriptions avoided. However, even in exploratory studies, it is possible and desirable to suggest practical implications and plausible interventions based on the results, always with some caution. For example, telemedicine programs, virtual support groups, or health literacy initiatives could be mentioned as potential strategies, especially in rural communities. We recommend including an "Implications for Practice" subsection with possible, data-based recommendations.

--Response: We agree. We have removed and rephrased all causal prescriptions. Implications are embedded throughout the discussion. We understand that data derived recommendations are invaluable, but in the context of our study, doing will be mostly speculation and not data driven. As noted by you, the cross-sectional data allows us only this much.

---

## [Decision Letter · Decision Letter 2]

7 Jan 2026

Antecedents of Loneliness Among Cancer Survivors: An Exploratory Analysis of the Health Information National Trends Survey (HINTS) Data.

PONE-D-25-18009R2

Dear Dr. Choudhury,

We’re pleased to inform you that your manuscript has been judged scientifically suitable for publication and will be formally accepted for publication once it meets all outstanding technical requirements.

Kind regards,

Taiwo Opeyemi Aremu, MD, MPH, PhD

Academic Editor

PLOS One

Additional Editor Comments (optional):

Reviewers' comments:

Reviewer's Responses to Questions

**Comments to the Author**

Reviewer #3: All comments have been addressed

2. Is the manuscript technically sound, and do the data support the conclusions?

Reviewer #3: Yes

3. Has the statistical analysis been performed appropriately and rigorously?

Reviewer #3: Yes

4. Have the authors made all data underlying the findings in their manuscript fully available?

Reviewer #3: No

5. Is the manuscript presented in an intelligible fashion and written in standard English?

Reviewer #3: Yes

Reviewer #3: All reviewer comments have been addressed, and the article is considered suitable for publication.

The authors are advised to verify all references and correct any remaining minor errors.

**Do you want your identity to be public for this peer review?** For information about this choice, including consent withdrawal, please see our Privacy Policy

Reviewer #3: No

---

## [Editor Report · Acceptance letter]

PONE-D-25-18009R2

PLOS One

Dear Dr. Choudhury,

I'm pleased to inform you that your manuscript has been deemed suitable for publication in PLOS One. Congratulations! Your manuscript is now being handed over to our production team.

Kind regards,

on behalf of

Dr. Taiwo Opeyemi Aremu

Academic Editor

PLOS One